# Development of a Transparent Transgenic Zebrafish Cellular Phenotype *Tg*(*6xNF-kB:EGFP); Casper*(*roy*^−/−^, *nacre*^−/−^) to Study NF-kB Activity

**DOI:** 10.3390/biomedicines11071985

**Published:** 2023-07-13

**Authors:** Surendra K. Rajpurohit, Logan Ouellette, Suvarsha Sura, Chelsea Appiah, Annabelle O’Keefe, Katherine McCarthy, Umasai Kandepu, May Ye Mon, Kirk Kimmerling, Vishal Arora, Bal L. Lokeshwar

**Affiliations:** 1Georgia Cancer Center, Medical College of Georgia, Augusta University, Augusta, GA 30912, USA; 2KHG FiteBac Technology, Marietta, GA 30064, USA; kirkkimmerling@fitebac.com; 3Division of Cardiology, Department of Medicine, Medical College of Georgia, Augusta University, Augusta, GA 30912, USA; varora@augusta.edu

**Keywords:** zebrafish, transgenic, transparent, casper, NF-kB, inflammation, in vivo imaging, fluorescent microscopy, confocal microscopy, cellular phenotype

## Abstract

NF-κB signaling has broad effects on cell survival, tissue growth, and proliferation activities. It controls many genes that are involved in inflammation and thus is a key player in many inflammatory diseases. The elevation of NF-κB activators is associated with elevated mortality, especially in cancer and cardiovascular diseases. The zebrafish has emerged as an important model for whole-organism in vivo modeling in translational research. In vertebrates, in-vivo spatial resolution is limited due to normal opacification of skin and subdermal structure. For in vivo imaging, skin transparency by blocking the pigmentation via chemical inhibition is required and the maintenance of this transparency is vital. The *Casper*(*roy*^−/−^, *nacre*^−/−^) mutant of zebrafish maintains this transparency throughout its life and serves as an ideal combination of sensitivity and resolution for in vivo stem cell analyses and imaging. We developed an NF-kB:GFP/Casper transparent transgenic zebrafish cellular phenotype to study inflammatory processes in vivo. We outline the experimental setup to generate a transparent transgenic *NF-kB*/*Casper* strain of zebrafish through the cross-breeding of *Casper* and NF-kB transgenic adult fish and have generated F01 in the form of heterozygous progeny. The transgenic F01 progeny was further inbred to generate heterozygous progenies from F1 to F4 generations. Furthermore, it continued to successfully develop the homozygous strain *Tg*(*6xNF-kB:EGFP); Casper*(*roy*^−/−^, *nacre*^−/−^) in the F05 generation. This novel strain of F05 generation showed 100% homozygosity in the transgenic transparent progeny of *Tg*(*6xNF-kB:EGFP)*; *Casper*(*roy*^−/−^, *nacre*^−/−^). The strain has been confirmed by generating the F06 generation of homozygous progeny and again verified and validated for its homogeneity in the F07 generation. The newly developed novel transparent transgenic strain of the NF-kB reporter line has been coined as “*Tg*(*6xNF-kB:EGFP)*; *Casper*(*roy*^−/−^, *nacre*^−/−^)*gmc1*”. We have established a newly generated phenotype of transparent transgenic zebrafish for time-lapse in vivo confocal microscopy to study the cellular phenotype and pathologies at the cellular level over time. This will allow for quantifying the changes in the NF-kB functional activities over time and allow the comparison of control and cardiac-oncology experimental therapeutics. We validated the newly developed *Tg*(*6xNF-kB:EGFP)*; *Casper*(*roy*^−/−^, *nacre*^−/−^)*gmc1* homozygous strain of zebrafish by studying the inflammatory response to bacterial lipopolysaccharide (LPS) exposure, tolerance, and the inhibitory role of a potential novel drug candidate against LPS-induced inflammation. The results establish the unique application of newly developed strains by identifying hit and lead drug candidates for experimental therapeutics.

## 1. Introduction

The zebrafish (*Danio rerio*) is an emerging vertebrate model organism for biomedical and translational research [1]. Zebrafish shares approximately 70% homology with humans in protein-coding and this semblance as humans allow for zebrafish to be utilized in understanding the progression of human diseases and treatment [2]. The zebrafish is a proven vertebrate model system for whole-organism-based drug discovery [3]. The novel characteristic of zebrafish is the transparency of eggs and embryos up to 12 h post-fertilization stage, and the average breeding pair has the capacity to produce and externally fertilize approximately 150–200 eggs per week [4]. We have developed and designed a custom-made mass eggs production device, which is capable of breeding an average of 100 pairs to produce approximately 10,000 eggs at a single time. In studies that involve vertebrates, a common issue for the in vivo spatial resolution is limited as the normal pigmentation of the epidermal layer and subdermal structures can lead to obstruction. The issue that pigmentation imposes can be alleviated using chemical melanization blockers, such as PTU (1-phenyl 2-thiourea) [5]. Chemical inhibition is a temporary method and is only able to maintain phenotype through continued treatment. Normal skin pigmentation of zebrafish demonstrates alternating patterns of deeply pigmentated strips composed, primarily, of melanocytes. The interstrip regions devoid of melanocytes contain reflective iridophores and yellowish xanthophores. The *nacre* mutant line, due to loss of mtf, shows a loss of embryonic and adult melanocytes with a compensatory increase in iridophore numbers. The *roy orbison* mutant line shows a striking abnormality of eye pigmentation, severe disruption of melanocyte numbers pattering, and a complete loss of the iridophore layer. The *Casper*(*roy*^−/−^, *nacre*^−/−^) mutant line shows the combined effect of melanophores loss and iridophore so the body of the adult fish is largely transparent through the entirety of its life and is an ideal combination for in vivo cell analysis and imaging [6]. *Casper* (*roy*^−/−^, *nacre*^−/−^) was the name coined by the founder Lab led by Dr L. Zon, from where we obtained the *Casper* mutant line for this experiment. The *Casper* mutant can be utilized as a non-invasive in vivo model system due to its high visibility of internal structures, allowing for real-time studies of whole-organ function in zebrafish. It is possible through the cross-breeding of the *Casper* line desired transgenic lines that use fluorescent reporter genes tagged to a cell or protein, allowing for a substantial number of studies to be conducted using this model.

Our laboratory is developing a transparent transgenic zebrafish modeling system to study cardio-oncology experimental therapeutics and research. We are using genetically engineered transgenic and mutant strains of zebrafish in multiple facets of our research niches ranging from zebrafish housing [7], technology and methodology design, development such as laser-induced thrombosis [8], and whole-organism screening platform enabling in vivo high-throughput rates drug screening [5]. Furthermore, this model has been used for human blood disorders [9,10] to study the unique signaling pathways controlling pancreatic β-cell neogenesis and mass, potential therapeutic targets for treating diabetes [11]. Recently, our laboratory has developed a novel transparent transgenic zebrafish strain *Tg*(*UAS:SEC-Hsa.ANXA5-YFP*, *myl7:RFP*); *Casper*(*roy*^−/−^, *nacre*^−/−^) to study annexin-5 activity in the cardiovascular function under normal and in metabolically aberrant conditions [12]. This is a newly developed transparent transgenic zebrafish strain to quantify changes in cardiomyocyte morphology and function through our established time-lapse in vivo confocal microscopy. Our model allows for specific studies to be conducted on diseases, cancer progression, or pharmacological interactions.

The NF-kB signaling pathway plays an immensely important role in physiological processes, such as inflammation, immune response, apoptosis, cell growth, and differentiation. Complications in this pathway have been seen to correlate with lethal diseases, such as cancer [13,14,15], cardiovascular health, and a host of other diseases [16,17]. In quiescent cells, the NF-κB complex is isolated to the cytoplasm and interacts with inhibitory IκB proteins. The activation of NF-κB signaling leads to kinase-dependent phosphorylation and the degradation of the IκB proteins causing NF-κB to be translocated to the nucleus and regulate target-gene transcription [18].

The zebrafish has unique conserved and compatible NF-kB/IkB proteins among vertebrates and shows the importance of the NF-kB pathway in mesoderm formation during early embryogenesis [19]. NF-κB signaling is implicated as a key node between cardiac injury and tissue regeneration [20]. The zebrafish transgenic pSGNluc has been established as a valuable tool to study many aspects of NF-κB signaling during development and inflammation in real time [21]. In our previous study, we have discovered the novel role of NF-κB signaling in regulating endocrine differentiation and serotonergic signaling in selectively stimulating β-cell proliferation [11]. Such strengths of our unique findings inspired us to study the role of NF-kB signaling in cardio-oncology. The previous study was conducted on the zebrafish larvae due to the limitation of tracking the cellular expression of normal skin pigmentation in the adult stage. To resolve this issue, we designed our experiment to develop a transparent transgenic strain to study NF-kB:GFP/Casper.

We have selected the transparent mutant *Casper*(*roy*^−/−^, *nacre*^−/−^) and the transgenic line NF-kB:GFP Tg(6xNFκB:EGFP). The NF-kB line has a green fluorescent protein (GFP) tagged to track the nuclear factor kappa light-chain-enhancer of activated B cells (NF-kB), which expresses inflammation when stimulated [22]. The developing line will express the transparent phenotype from the *Casper* mutant and the genotype GFP from the transgenic NF-kB line. The experiment has been conducted in two stages: (1) generating the heterozygous and homozygous lines through cross-breeding NF-kB:GFP Tg(6xNFκB:EGFP) and *Casper* (*roy*^−/−^, *nacre*^−/−^) (generations F01-F07); (2) screening and sorting the transparent transgenic progeny using in vivo imaging to validate genotypical expression. Confocal and fluorescent microscopy were used to image and determine the fluorescent protein being activated. We named the newly developed strain *Tg*(*6xNFκB:EGFP*)*; Casper*(*roy*^−/−^, *nacre*^−/−^)*^gmc1^* and used *Casper: NF-kB:GFP* as the running name.

The newly developed *Tg*(*6xNF-kB:EGFP); Casper*(*roy*^−/−^, *nacre*^−/−^) strain of zebrafish has been validated by studying the inflammatory response to bacterial lipopolysaccharide (LPS) exposure, tolerance, and the inhibitory role of a potential novel drug candidate against LPS-induced inflammation. We have used the K21 drug candidate to evaluate its anti-inflammatory or inhibitory role against inflammations and established the unique application of newly developed strains by identifying hit and lead drug candidates for experimental therapeutics.

This transgenic line could expedite the development of treatment for cardiovascular diseases and cancer. Our approach could yield crucial new insights into in vivo cardiomyocyte imaging via confocal microscopy. Tracking the cellular inflammation pattern in the cellular phenotype and microglia, the resident macrophages of the brain, can lead to the development of novel therapeutic approaches. The zebrafish model will allow for insights into the in vivo imaging via a confocal microscope and is capable of tracking cell death patterns in the NF-kB pathway to develop novel therapeutics.

## 2. Methods and Materials

### 2.1. Fish Housing

The zebrafish strains used in this experiment are maintained in the Augusta University Transgenic Zebrafish Core Facility (TZF Core). This facility maintains a controlled environment of water sterility to optimal biological conditions and water filtration through pH, temperature, conductivity, and UV lights. A constant 14:10 h light–dark cycle was maintained, and the temperature was kept between 27.0 and 28.5 °C for the embryos, larvae, and adult zebrafish. A controlled feeding regimen is maintained to instigate robust mating. The Zebrafish Core Laboratory sustains the various zebrafish strains in appropriately sized tanks regarding size and population. The various strains of zebrafish and their stage for this experimental design and procedures were selected in accordance with the agreed protocol standards from Augusta University Animal Care and Use Committees (IACUCs).

### 2.2. Zebrafish Line

In this experiment, we have used the following strains of zebrafish: wild type, WT-AB; transgenic lines, NF-kB:GFP *Tg*(*6xNFκB:EGFP*) [22]; and transparent skin mutant line “*Casper* (*roy*^−/−^, *nacre*^−/−^) [6]”. The fish line *Casper*(*roy*^−/−^, *nacre*^−/−^) mutant was generously gifted from Dr Zon’s lab following its development from the cross-breeding between two developed mutant lines, *roy^−/−^* and *nacre*^−/−^. The NF-kB reporter transgenic line Tg(6xNF-κB:EGFP) fish was purchased from ZIRC (Zebrafish International Resource Center).

### 2.3. Equipment

#### 2.3.1. Fluorescent Microscope

To confirm transgenic expression, we used a Keyece fluorescence microscope (model BZX-800) Revolve model of ECHO: A BICO Company, which is capable of performing under light conditions. This was used on larvae at the 72 hpf (hours post-fertilization) stage.

#### 2.3.2. Confocal Microscope

To obtain in vivo imaging of the zebrafish larvae at 72 hpf, 96 hpf, and at the adult stage, we primarily used a thorough Leica Confocal microscope (model: Leica Stellaris-5 Microsystem). We have also used the ZESIS Axioplan2 model and ZESIS LCM 780 Multiphoton Confocal microscope for confocal microscopy as needed. The images were captured and analyzed using LAS-X software (Version 4.6.1.27508; Leica Microsystems, Belgrade, Serbia).

#### 2.3.3. Screening Imaging via Fluorescence and Confocal Microscopy In Vivo

The in vivo time-lapse imaging that was established for the confocal microscopy was used to observe the cellular pathologies through the affected tissues. The fish returned to controlled conditions following the imagery. Image Analysis: Using the established ImageJ and TIFF file the images were analyzed for cellular morphological studies. Our common fluorescent is expressed through the following emission and excitations, respectively: green fluorescent protein (GFP): excitation, 488 nm [2] and emission, 507 nm; yellow fluorescent protein (YFP): excitation, 520 nm and emission, 546 nm [5,11,12].

### 2.4. Chemicals and Reagents

#### 2.4.1. Fish Water

The Zebrafish Core Facility Water Circulating System supplies the fish water for fish units. The commonly used 1× E3 media was evaluated and compared to the fish water before implementation. To prepare the 1× E3 media, a combination of 29.22 g of NaCl, 1.27 g of KCl, 3.33 g of CaCl_2_, and 3.97 g of MgSO_4_ was used to produce 1 L of 100× E3 medium. The medium was placed on a stir plate at room temperature (RT: 21 °C) for 1 h. The pH was adjusted to 7.4 with NaOH. Stocks of 100× E3 were stored at RT for 3 months or at 4 °C for 6 months. To prepare the working 1× E3 medium, the 100× E3 medium was diluted (above) in water to achieve a 1× concentration. The final 1× E3 solute concentration was as follows: NaCl (5 mM), KCl (0.17 mM), CaCl_2_ (0.3 mM), and MgSO_4_ (0.33 mM). Stocks of 1× E3 were stored at RT (room temperature) for 3 months [12]. The comparison concluded that the TZF Core produced water was adequate and more cost-efficient for the maintenance of the zebrafish colony.

#### 2.4.2. Phenyl-2-Thiourea (PTU)

Phenyl-2-thiourea (PTU) (Acros Organics (cat no. 103-85-5)) when handling PPE in accordance with its high toxicity is worn. A 50× PTU stock solution is prepared by combining 1.520 g of PTU with every 1 L of 1× E3 medium. The solution is mixed on a hot plate for 2 h at 50 °C. The produced solution cooled to RT, and aliquots were made in the fume hood. The aliquoted PTU stock is sustainable for 1 year when stored at −20 °C (protected from light). The 50× PTU solution was diluted to 1× with 1× E3, and the diluted solution is capable of being stored at RT for up to 3 months [22]. PTU is used as a tyrosinase inhibitor to block melanization. The zebrafish embryos begin to develop melanophores in the skin from 14 hpf, which persist throughout their life. Fluorescent images are taken on the 72 hpf mark with the larvae treated with the PTU at a concentration of 0.003% embryonic media (EM) at 12 hpf to block the expression of skin pigmentation [12].

#### 2.4.3. Stress/Pain Management

Clover oil was included in the working solution to alleviate stress and/or pain-induced movement for screening through a microscope and imaging by confocal microscopy. To further alleviate stress during the fluorescent and confocal microscopy, protocols were put into place, including, and not exclusive to, dimmed lighting and reduced noise [5,12].

### 2.5. Experiment Design

#### 2.5.1. Generation of the Heterozygous and Homozygous Progeny *Casper* (*roy*^−/−^, *nacre*^−/−^)*/NF-kB:GFP* (Generations F01–F07)

The zebrafish take 3 months of development per generation to develop into the fertile adult stage [4]. The development of new strains, from F01 generation, through the cross-breeding of an individual *Casper* mutant with a transgenic strain of *NF-kB:GFP* to produce the F01 progeny (Figure 1). The transgenic F01 progeny was inbred to generate F02–F04 generations (heterozygous). The continued inbreeding produced the F05 to F07 generation, which achieved the production of the homozygous strain. The breeding procedure during a maintained breeding cycle can take a 24-month period.

Breeding setup (Day Zero): The male and female zebrafish from the selected strains were paired and placed in a breeding tank for matting.

Day 1 (0–24 hpf): The produced embryos were collected from the bottom of the breeding tanks, debris was removed, they were transported to deep-dish Petri dishes, and given time to grow.

Day 1: (12 hpf): The embryos are cleaned, and the dead embryos are removed. The remaining embryos are treated with PTU which blocks pigmentation.

Day 3 (49–72 hpf): The embryos are observed to have hatched into larvae. At this stage, the larvae were screened for their transgenic expression via fluorescent microscopy, and the transgenic and non-transgenic progeny were separated.

For the further development of the zebrafish post-screening, the larvae were transferred into fresh TZF Core system water.

Day 6 to 9 (6–9 dpf): The larvae feeding starts at the 6–9 days post-fertilization (dpf) and there is maintained cleaning.

Day 10 (10 dpf): The larvae are transferred into regular fish tanks and placed on the assigned larvae development racks of the zebrafish core facility. A diet of hatched brine shrimp is fed from 10 to 12 dpf, after which normal feeding occurs.

Day 12+ (12+ dpf): The new progeny is developed into the adulthood stage in the period of three months.

The adult transgenic heterozygous progeny from the F01 generation was selected for breeding, and this was maintained in the following generations (F02–F04) to create a homozygous strain (F05–F07).

#### 2.5.2. Screening of Fluorescent Expression to Select Transgenic Phenotype through Fluorescent Microscopy

A model BZX-800 fluorescent microscope was used to screen the zebrafish larvae at the 72 hpf stage [12].

The larvae of the newly developed strain *Tg*(*NF-kB:GFP*)*; Casper*(*roy*^−/−^, *nacre*^−/−^) were obtained at the 72 hpf stage and selected. Then, they were screened for GFP expression using fluorescent microscopy to observe the fluorescent reports [12].

The selected larvae were divided into a transgenic group Tg(NF-kB:GFP) with green-fluorescent expression and a non-transgenic group without fluorescence.

In vivo confocal imaging of 12 transgenic larvae from each positive transgenic group was carried out for further confirmation and validation purposes.

The progeny of transgenic and non-transgenic generation larvae was transferred to separate tanks, properly labeled, and allowed to grow into the adult stage for the next-generation breeding.

#### 2.5.3. Confirmation and Validation of In Vivo Imaging of Homozygous Strain through Confocal Microscopy

Confocal microscopy was carried out using the ZESIS Axioplan2 and ZESIS LCM 780 Multiphoton Confocal microscope models on progeny at the following stages: 48 hpf, 72 hpf, 96 hpf, and adult.

The larvae of the newly developed strain of Tg(NF-kB:GFP); Casper(roy^−/−^, nacre^−/−^) were selected at the 72 hpf stage and screened for their fluorescence [12].

The selected larvae were mounted in 1.6% low-melt agarose. The prepared agarose is continually maintained at 35 °C.

A sample of the produced progeny was anesthetized in a solution containing 0.5 mL of 0.02% working media clove oil and 9.5 mL of TZF Core system water and transferred to a 1.5 mL Eppendorf tube.

The individually anesthetized fish from the studied progeny and stage was added to 50 µL of the 1.6% agarose solution, and the progeny was gently mixed and immediately poured onto the glass-bottom microwell Petri dish (Mat Tek) (note: the zebrafish usually position themselves flat on their side, though some required gentle reorientation with a pipette tip as the agarose solidified). The progeny being fixed allows for easy observation under fluorescent microscopy and for the time-lapse in vivo confocal imaging. The imagery was processed in Z-stacking at the interval of 5 µ of the whole organism covering the fluorescence expression at the tissue level.

#### 2.5.4. Flow Cytometry Analysis for Validation of the Newly Developed Zebrafish Strain

Tg(6xNF-kB:EGFP); Casper(roy^−/−^, nacre^−/−^) by evaluating the inflammatory response to bacterial lipopolysaccharide (LPS) exposure, and the inhibitory role of a potential novel drug candidate against LPS-induced inflammation.

For this experimental setup, larvae of Casper (transparent mutant) as control and newly developed *Casper: NF-kB:GFP* transgenic progenies were used. Three days old larvae of both groups were treated with system water (negative control), LPS, and K21 drug. These larvae were treated for 4 days. The LPS was used in a concentration of 100 μg/mL dissolved in the system water, while the K21 drug dose was optimized and used in a concentration of 10 μg/mL. The experiment was conducted by forming 4 groups of treatment regimens:Group 1: Control only fish system water (Casper mutant and *Casper: NF-kB:GFP*)Group 2: LPS treatment fish system water (Casper mutant and *Casper: NF-kB:GFP*)Group 3: LPS + K21 Drug treatment fish system water (Casper mutant and *Casper: NF-kB:GFP*)Group 4: K21 Drug treatment fish system water (Casper mutant and *Casper: NF-kB:GFP*)

Larval dispensing in a 96-well plate: Twenty-four larvae were used for each individual condition. Each larva was dispensed into an individual well of the 96-well plate containing 300 μL of media.

The larvae were treated for 4 days. The plate was covered by a lid to avoid evaporation. A temperature of 28 °C was maintained throughout the duration of the experiment.

Flow cytometry Assay: After 4 days of treatment, the whole larvae were homogenized using a glass homogenizer (Pyrex potter tissue grinder; cat no Pyrex 7525T3), and the supernatant was filtered using a 70 μm nylon mesh sterile cell strainer (Fisher; cat no. 22363548). The fresh cell samples were analyzed for flow cytometry analysis at the AU Flow Cytometry Core using the Cytec Biosciences Aurora Spectral Flow cytometer model.

## 3. Results

### 3.1. Generation of Tg(6xNF-kB:EGFP); Casper(roy^−/−^, nacre^−/−^) Strain

F01 Generation: To develop the proposed strain, we selected the transparent mutant Casper and NF-kB:GFP reporter transgenic line Tg(6xNF-kB:EGFP) fish strains. These strains were used to generate the new strain. A set of six breeding pairs of male Casper (roy^−/−^, nacre^−/−^) and female *Tg*(*6xNF-kB:EGFP*) fish were crossbred to produce the F01 progeny. Approximately 80% of the eggs fertilized and developed into embryos. The developed embryos were treated with PTU to block the pigmentation at the 10 hpf stage. The larvae were developed to the 72 hpf stage. At the 72 hpf stage, the larvae were screened for EGPF expression; of the screened F01 progeny, 39% expressed GFP, while 61% lacked the GFP (Figure 2). The screened larvae were developed into the adult stage. As the screened larvae were developed to the adult stage, they expressed 100% WT-AB pigmentation with regard to phenotypical expression (Figure 3, Figure 5 and Figure 6).

F02 Generation: The developed positively expressing Tg(6xNF-kB:EGFP) transgenic lineage was paired and inbred to produce the F02 generation. The established screening procedure at the 72 hpf stage expressed 79% positive GFP transgenic expression and 21% non-transgenic expression of the total larval population produced. The F02 generation followed the established development procedures and had the phenotype recorded at the 3-month adult stage. The total phenotypical rates of the F02 generation expressed were as follows: Casper(roy^−/−^, nacre^−/−^) 0%, nacre^−/−^ mutant 8%, roy^−/−^ orbison mutant 17%, and the WT-AB 75% (Figure 3, Figure 4, Figure 5 and Figure 6). The mixed expression of the roy^−/−^ and nacre^−/−^ mutants is caused by the developmental procedure of the Casper(roy^−/−^, nacre^−/−^) mutation, where the individual lines can be expressed without any developmental degeneration [6].

F03 Generation: The developed F02 adult transgenic progeny expressing GFP was selected as an inbred for the F03 generation. The established screening procedure at the 72 hpf stage expressed 72% positive GFP transgenic expression and 28% non-transgenic expression of the total larval population produced. The F03 generation followed the established development procedures and had the phenotype recorded at the 3-month adult stage. The total phenotypical rates of the F03 generation expressed were as follows: Casper(roy^−/−^,nacre^−/−^) 6%, nacre^−/−^ mutant 21%, roy^−/−^ orbison mutant 11%, and the WT-AB 63% (Figure 3, Figure 4, Figure 5 and Figure 6).

F04 Generation: The developed F03 adult transparent transgenic progeny expressing GFP was selected as an inbred for the F04 generation. The established screening procedure at the 72 hpf stage expressed 88% positive GFP transgenic expression and 12% non-transgenic expression of the total larval population produced. The F04 generation followed the established development procedures and had the phenotype recorded at the 3-month adult stage. The total phenotypical rates of the F03 generation expressed were as follows: Casper(roy^−/−^,nacre^−/−^) 88%, nacre^−/−^ mutant 0%, roy^−/−^ orbison mutant 0%, and the WT-AB 12% (Figure 3, Figure 4, Figure 5 and Figure 6).

F05 Generation: The developed F04 adult transparent transgenic progeny expressing GFP was selected as an inbred for the F05 generation. The established screening procedure at the 72 hpf stage expressed 100% positive GFP transgenic expression and 0% non-transgenic expression of the total larval population produced. The F05 generation followed the established development procedures and had the phenotype recorded at the 3-month adult stage. The total phenotypical rates of the F03 generation expressed were as follows: Casper(roy^−/−^,nacre^−/−^) 100%, nacre^−/−^ mutant 0%, roy^−/−^ orbison mutant 0%, and the WT-AB 0% (Figure 3, Figure 4, Figure 5 and Figure 6).

F06 Generation (Homozygous): The developed F05 adult progeny was selected as an inbred for the F06 generation. The established screening procedure at the 72 hpf stage expressed 100% positive GFP transgenic expression and 0% non-transgenic expression of the total larval population produced. The F06 generation followed the established development procedures and had the phenotype recorded at the 3-month adult stage. The total phenotypical rates of the F03 generation expressed were as follows: Casper(roy^−/−^,nacre^−/−^) 100%, nacre^−/−^ mutant 0%, roy^−/−^ orbison mutant 0%, and the WT-AB 0% (Figure 3, Figure 4, Figure 5 and Figure 6). The F06 generation confirmed the creation of a homozygous lineage.

F07 Generation (Homozygous): The developed F06 adult progeny was selected as an inbred for the F07 generation. The established screening procedure at the 72 hpf stage expressed 100% positive GFP transgenic expression and 0% non-transgenic expression of the total larval population produced. The F07 generation followed the established development procedures and had the phenotype recorded at the 3-month adult stage. The total phenotypical rates of the F03 generation expressed were as follows: Casper(roy^−/−^,nacre^−/−^) 100%, nacre^−/−^ mutant 0%, roy^−/−^ orbison mutant 0%, and the WT-AB 0% (Figure 3, Figure 4, Figure 5 and Figure 6). The F07 generation validated the development of the homozygous line and officially received the lineage *Tg*(*6xNF-kB:EGFP); Casper*(*roy*^−/−^, *nacre*^−/−^) (Table 1).

### 3.2. Screening and Sorting the Transgenic Progeny and In Vivo Imaging to Validate NF-kB Activity through Confocal Imaging of Tg(6xNF-kB:EGFP); Casper(roy^−/−^, nacre^−/−^) Strain

We have established fluorescence microscopy and in vivo time-lapse confocal microscopy methodology to screen larvae to detect and observe transgenic expression [12]. The newly generated larvae at the 72 hpf stage from each generation were screened through a fluorescent microscope (Keyence BZX-800 and Revolve model of ECHO). The embryos of the heterogeneous progeny of *Tg*(*6xNF-kB:EGFP*)*; Casper*(*roy*^−/−^, *nacre*^−/−^) strain from F01 to F04 generations were treated with PTU to block the pigmentation to maintain the transparency for fluorescence microscopy. The F05 generation progeny was observed as homozygous in nature having a 100% transparent skin background in the adult population. The same steps were followed for the F6 and F7 generation GFP transgenic expression screening. For validation purposes, the confocal microscopy was conducted at various development stages, such as 48 HPF, 72 HPF, 96 HPS and adult stage, of the newly developed *Tg*(*6xNF-kB:EGFP*)*; Casper*(*roy*^−/−^, *nacre*^−/−^) strain (Figure 7).

### 3.3. Validation of the Casper; NF-kB:GFP Strain by Evaluating the Inflammatory Response to LPS Exposure, and the Inhibitory Role of a Potential Novel Drug Candidate against LPS-Induced Inflammation

For further validation of the newly developed NF-kB:GFP Tg(6xNF-kB:EGFP); Casper(roy^−/−^, nacre^−/−^) zebrafish strain, we have conducted an additional experiment to evaluate the inflammatory response to bacterial lipopolysaccharide (LPS) exposure. We have also studied the novel drug candidate K21 to evaluate its potential inhibitory role against LPS-induced inflammation in the newly developed zebrafish strain.

In the transparent transgenic Casper/NF-kB:GFP larvae, the results show the impact of LPS-induced inflammation by significant over-expression of GFP (9.79%) in the LPS-treated experimental group as compared to the control group treated with system water (4.34%), while the LPS + K21 drug group reduces the GFP expression by 7.08% and only K21 drug group shows even lower than the control group (4.05%) (Figure 8). To rule out any artifact or false observation, we used the larvae of non-transgenic casper mutant fish in this experiment. We did not observe any GFP expression in any of the counterparts’ groups. These novel findings established the inflammatory-response-induced LPS treatment and the inhibitory effect of the K21 drug on LPS-induced inflammation.

## 4. Discussion

NF-kB is an important transcription factor which regulates a wide range of cellular processes and pathways. Inflammation is a complex physiological process. Like mammals, zebrafish also have five members in the family of NF-κB transcription factors. Dr Rawl’s Lab characterized the NF-κB reporter transgenic line *Tg*(*6xNFκB:EGFP*) in zebrafish [22]. We have successfully used this NF-kB transgenic line in our previous study, “First quantitative high-throughput screen in zebrafish to identify novel pathways for increasing pancreatic β-cell mass”, and we identified 11 lead drugs, in which 2 drugs, thioctic acid and parthenolide, are known inhibitors of the NF-κB signaling pathway [11].

Our laboratory is developing a transparent transgenic zebrafish modeling system for studying experimental therapeutics in the field of cardio-oncology. We have designed and established an in vivo time-series state-of-the-art confocal microscopy imaging methodology, which allows tracking cellular phenotype/pathologies over time in individual zebrafish strains by developing Tg(UAS:SEC-it’s the. ANXA5-YFP,myl7:RFP); Casper(roy^−/−^, nacre^−/−^) transgenic zebrafish strain [12]. In the present study, we have developed a new strain of stable transparent mutant transgenic NF-kB reporter line *Tg*(*6xNF-kB:EGFP*)*; Casper*(*roy*^−/−^, *nacre*^−/−^).

Transgenic zebrafish have normal-type skin pigmentation background. Normal skin pigmentation of zebrafish demonstrates an alternating pattern of deeply pigmentated strips composed mainly of melanocytes. To maintain transparency throughout the life of the NF-kB transgenic reporter line, we designed this experiment by crossbreeding the *Tg*(*6xNF-kB:EGFP* strain with the skin transparent mutant Casper(roy^−/−^, nacre^−/−^). We developed a new strain of stable transparent mutant transgenic NF-kB reporter line *Tg*(*6xNF-kB:EGFP*)*; Casper*(*roy*^−/−^, *nacre*^−/−^). By developing this novel strain, we established time-lapse in vivo confocal microscopy to study cellular phenotype/pathologies (Figure 3, Figure 4, Figure 6 and Figure 7).

The newly developed NF-kB:GFP Tg(6xNF-kB:EGFP); Casper(roy^−/−^, nacre^−/−^) zebrafish strain has been further validated by conducting an additional experiment to evaluate the inflammatory response to bacterial lipopolysaccharide (LPS) exposure by performing flow cytometry analysis using whole larvae. We have also studied the novel drug candidate K21 to evaluate its potential inhibitory role against LPS-induced inflammation in the newly developed zebrafish strain. K21 is a broad-spectrum antimicrobial for the treatment of a wide variety of diseases. In our laboratory, we are conducting a study to evaluate the impact of the K21 drug using a zebrafish modeling system, evaluating its toxicity and biomedical and therapeutic application. These novel findings established the inflammatory-response-induced LPS treatment and the inhibitory effect of the K21 drug on LPS-induced inflammation.

## 5. Conclusions

We have successfully developed a novel transgenic homozygous strain *Tg*(*6xNF-kB:EGFP)*; *Casper*(*roy*^−/−^, *nacre*^−/−^) in the F05 generation. This novel strain of the F05 generation is 100% homozygous in the transgenic transparent progeny of *Tg*(*6xNF-kB:EGFP)*; *Casper*(*roy*^−/−^, *nacre*^−/−^). The strain was confirmed and validated by the F06 and F07 generations, respectively. The newly developed novel transparent transgenic strain of NF-kB reporter line has been termed “*Tg*(*6xNF-kB:EGFP)*; *Casper*(*roy*^−/−^, *nacre*^−/−^)*gmc1*”. We have used the K21 drug candidate to evaluate its anti-inflammatory or inhibitory role against inflammations and established the unique application of newly developed strains by identifying hit and lead drug candidates for experimental therapeutics. This novel transgenic model will be utilized in the study of FDA-approved anti-inflammatory drug inhibitors in NF-kB pathophysiology for cardio-oncology experimental therapeutics.

## Figures and Tables

**Figure 1 biomedicines-11-01985-f001:**
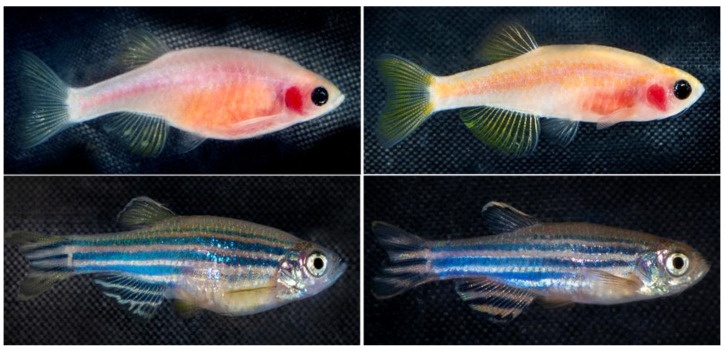
The zebrafish phenotype expression used in this experiment is Casper(roy^−/−^, nacre^−/−^) (**top row**) and the WT-AB phenotype (**bottom row**). An example of each sex is present for each phenotype, the females for each line (**left column**) and the males (**right column**). The inflammatory transgenic Tg(NF-kB:GFP) has a normal wild-type skin pigmentation pattern.

**Figure 2 biomedicines-11-01985-f002:**
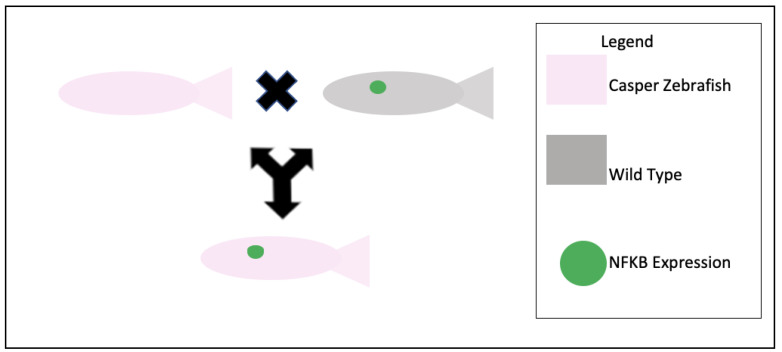
Cross-breeding of Casper and NF-kB:EGFP zebrafish progeny. The green dot represents the presence of NF-kb:GFP. The experiment is to develop a transparent transgenic zebrafish that expresses both the Casper phenotype and the NF-kB genotype.

**Figure 3 biomedicines-11-01985-f003:**
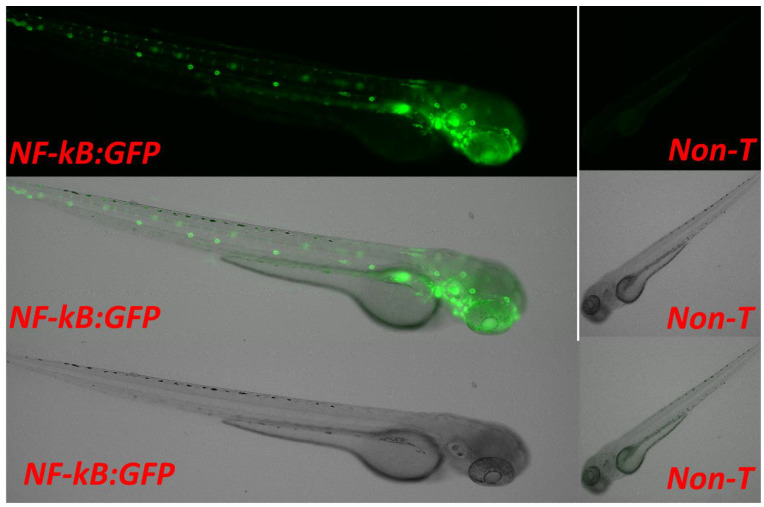
In vivo imaging of the 72 hpf (hours post-fertilization) stage of the newly developed zebrafish strain of Tg(NF-kB:EGFP); Casper(roy^−/−^, nacre^−/−^). Green fluorescent protein (EGFP) shows NF-kB specific to inflammatory protein and Casper transparent skin mutant in which they represent deleted melanophore and xanthophore genes that make the transparent skin.

**Figure 4 biomedicines-11-01985-f004:**
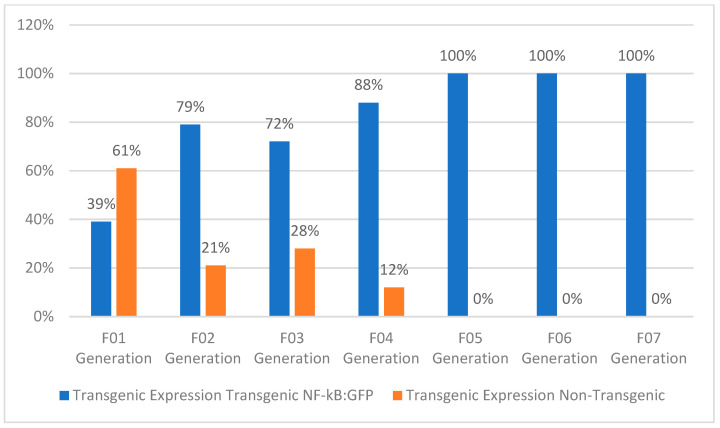
Transgenic expression in newly generated zebrafish cellular phenotype of *Tg*(*NF-kB:EGFP); Casper*(*roy*^−/−^, *nacre*^−/−^) (percentage of population).

**Figure 5 biomedicines-11-01985-f005:**
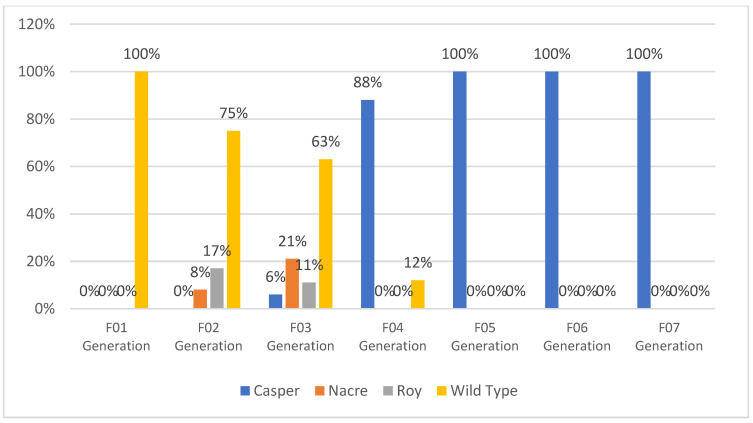
Percentile of skin pigmentation background pattern of newly generated NF-kB:EGFP zebrafish phenotype, the *Casper*(*roy*^−/−^, *nacre*^−/−^), the *nacre* mutant, the *roy orbison* mutant, and the WT-AB normal skin pigmentation pattern at the adult stage (percentage of population).

**Figure 6 biomedicines-11-01985-f006:**
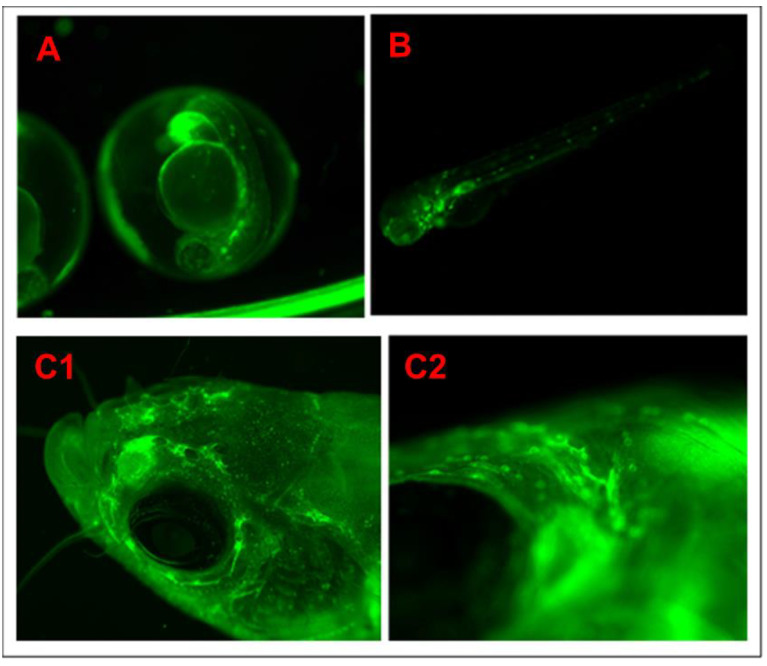
The green fluorescent protein (EGFP) shows NF-kB specific expression and the *Casper* transparent skin mutant in which they represent deleted melanophore and xanthophore genes that make the transparent skin. (**A**) Embryonic stage at 42 hpf (hours post-fertilization), (**B**) larval stage at 96 hpf (hours post-fertilization), and (**C1**,**C2**) head region of the adult stage, (**C1**): 1.5× image and (**C2**): 7× image.

**Figure 7 biomedicines-11-01985-f007:**
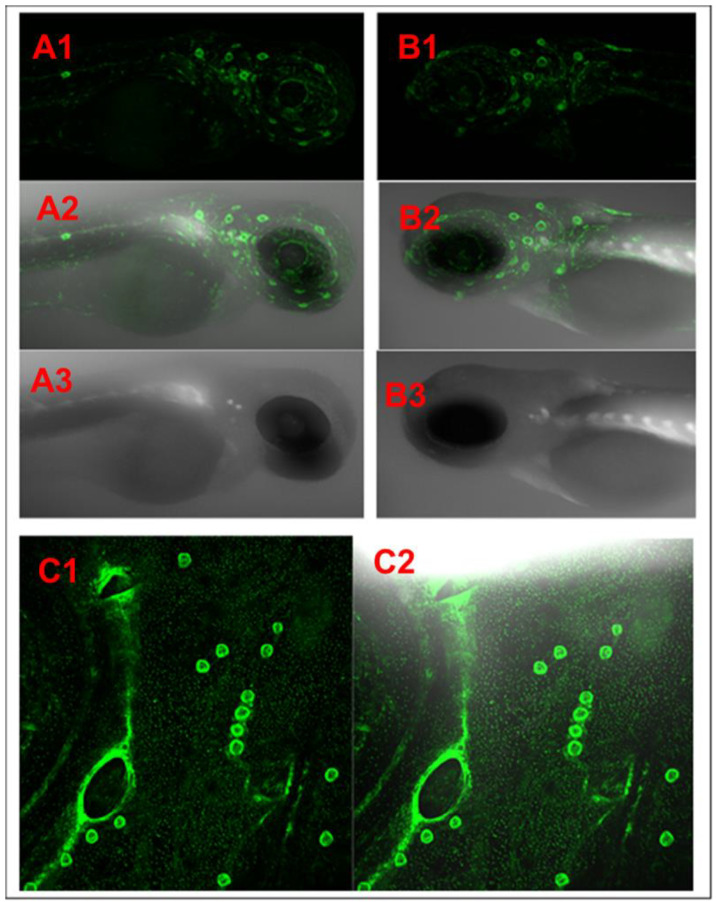
Confocal in vivo imaging of the newly developed zebrafish strain *Tg*(*NF-kB:EGFP); Casper*(*roy*^−/−^, *nacre*^−/−^). The green fluorescent protein (EGFP) shows NF-kB specific expression and the *Casper* transparent skin mutant in which they represent deleted melanophore and xanthophore genes that make the transparent skin. (**A1**–**A3**) Larval stage at 72 hpf (hours post-fertilization), (**B1**–**B3**) larval stage at 96 hpf (hours post-fertilization), and (**C1**,**C2**) head region of the adult stage.

**Figure 8 biomedicines-11-01985-f008:**
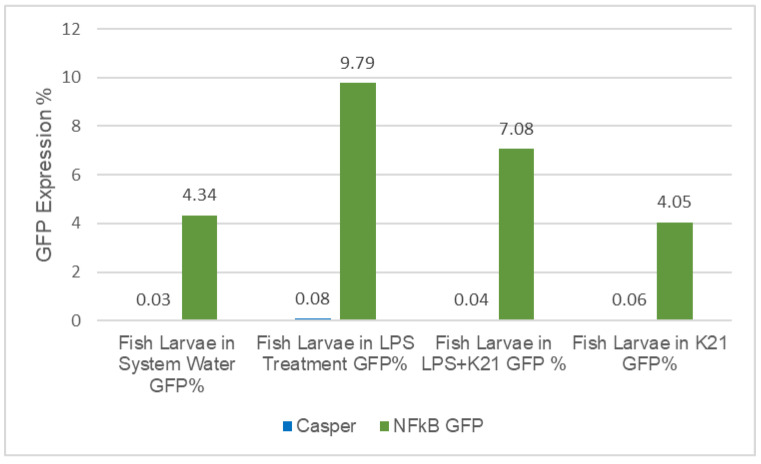
Flow cytometry Analysis: The inflammatory impact of LPS treatment and the inhibitory effect of K21 drug in fish larvae of Casper mutant (blue bar) as control and GFP expression in the transparent transgenic Casper/NF-kB:GFP strain (green bar). The GFP expression of flow cytometry data expressed in percentile (%).

**Table 1 biomedicines-11-01985-t001:** Summary of the newly developed strain Tg (6xNF-kB:EGFP); Casper (roy^−/−^, nacre^−/−^).

	Progeny	Strain	Transgenic Expression		Phenotypically Skin Pigmentation Background			
	Generation	Heterozygous	Transgenic	Non-Transgenic	Casper	Nacre	Roy	Wild Type
		Homozygous	NF-kB:GFP					Normal Skin
1	F01	Heterozygous	39%	61%	0%	0%	0%	100%
2	F02	Heterozygous	79%	21%	0%	8%	17%	75%
3	F03	Heterozygous	75%	25%	6%	21%	11%	63%
4	F04	Heterozygous	100%	0%	88%	12%	0%	0%
5	F05	Homozygous	100%	0%	100%	0%	0%	0%
6	F06	Homozygous	100%	0%	100%	0%	0%	0%
7	F07	Homozygous	100%	0%	100%	0%	0%	0%

## Data Availability

The original data presented in this study can be obtained from authors upon request.

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
