# Peer review of "Development of a Transparent Transgenic Zebrafish Cellular Phenotype Tg(6xNF-kB:EGFP); Casper(roy−/−, nacre−/−) to Study NF-kB Activity"

_biomedicines, 2023, doi:10.3390/biomedicines11071985_

Round 1

Reviewer 1 Report

The manuscript deals with an interesting topic and is well organized. However, the manuscript is written in a confusing way with grammar errors and punctuation that needs to be revised.

Furthermore, the authors emphasize in the introduction and discussion their previous work in which they used Casper(roy−/−,nacre−/−) transgenic zebrafish strain to study cardiovascular diseases, but fail well to explain the connection with the new model proposed here and how this new zebrafish strain could be useful in the study of these or other pathologies. 

Finally, the discussion is poor and only summarizes the result obtained without any consideration or real discussion on the usefulness of this strain and therefore of this work. Moreover, the authors have to specify the abbreviations when reported for the first time. 

Furthermore, the authors have to control grammar and spelling errors and punctuation marks to make more simple understand the manuscript, especially in Materials and Methods section. 

In the lines  294-297 what are indicating 3.1 and 3.2? 

To conclude, the manuscript needs to be rewritten better with more attention to be ready for publication.

The English need to be revised since grammar and spelling errors are present.

Author Response

Reply attached

Reviewer 2 Report

The manuscript by Rajpurohit and collaborators entitled “Development of Transparent Transgenic Zebrafish Cellular Phenotype Tg(6xNF-kB:EGFP; Casper(roy−/−, nacre−/−) to Study NF-kB Activity” described the development of a new zebrafish model where a reporter gene allows the labeling of the NF-kB expressing cells in zebrafish in order to get a tool for future pharmacological or cancer-related studies.

Although the work in itself could be useful for the scientific community, and it took a lot of work to get the final model, I think that this paper could not be accepted in its current form.

My major concerns are related to the absence of a more accurate identification of the specific pattern of expression of NFkB, for example by identifying inflammatory cells at the larval stage; a co-staining for LysC or mpeg1, etc could be helpful to mark NFkB:EGFP positive leukocytes at this stage. The same consideration is true for cardiac cells.

The authors talked about future functional studies, but they should provide here experimental data about the successful use of the new model.  

As minor issue, but at one point the become important for the complete understanding of the work, the authors should carefully revise the captions, avoid repeated sentences and accurately label each image, with the corresponding description in the text.

For figure 2, I understand the meaning of the dot – color legend, but I would rather use a different pattern to avoid confusion and maybe let the reader think about a specific localization.

Globally, the manuscript text would need to be improved; please avoid repeating the same sentences all over the text.

Author Response

Reply to Reviewer-2 comments attached

Round 2

Reviewer 1 Report

The authors have improved the Manuscript as I requested.